# Multimodality Treatment and Salvage Surgery for the Treatment of Lung Cancer

**DOI:** 10.3390/cancers15143586

**Published:** 2023-07-12

**Authors:** Jeesoo Choi, Boris Tocco, Alexander Smith, Shahreen Ahmad, Eleni Josephides, Andrea Bille

**Affiliations:** 1Thoracic Surgery, Guy’s Hospital, London SE1 9RT, UK; 2Department of Clinical Oncology, Guy’s Hospital, London SE1 9RT, UK; boris.tocco@gstt.nhs.uk (B.T.); eleni.josephides@kcl.ac.uk (E.J.)

**Keywords:** lung cancer, trimodality, salvage

## Abstract

**Simple Summary:**

There is a broad variety of treatment options for lung cancer, which include medical therapy and surgical resection. Opinion varies as to the optimal treatment options and combinations in different contexts of the disease, which has an impact on the prognosis of the disease—relapse and survival. This is a descriptive study on patients who underwent multimodality therapy (a combination of chemotherapy, radiotherapy and surgical resection) and an evaluation of patient and disease characteristics that may have an impact on the treatment pathway.

**Abstract:**

Lung cancer remains the leading cause of cancer deaths in the United Kingdom. For locally advanced disease, multimodality treatment is recommended, which includes a combination of chemotherapy, radiotherapy, surgery and, more recently immunotherapy. Options depend on the resectability of the cancer and there has been debate about the optimal treatment strategy: surgery may be planned to follow chemoradiotherapy (CRT), be offered for residual disease after CRT, or given as salvage therapy for patients treated with CRT who have later relapse of their disease. We conducted a retrospective analysis of all patients who underwent CRT and surgical resection under a single surgical team and performed a descriptive study after dividing the patients into these three groups. For the planned trimodality group, 30-day mortality this was 7% (*n* = 1) and 1-year survival was 78.6%; the residual disease group had a 30-day mortality rate of 0% and 1-year survival of 81.3%; for the salvage group, the figures were 0% and 62.5%, respectively. The median overall survival of the study population was 35.8 months. Median overall survival in the trimodality group was 35.4 months (20.1–51.7 interquartile range IQR), for the residual group was 34.2 months (18.5–61.0 IQR). and for the salvage group was 35.8 months (32.4–52.7 IQR).)

## 1. Introduction

A recent study published by the International Association for the Study of Lung Cancer showed that the number of people with lung cancer diagnoses in the United Kingdom was more than 47,000, and it continues to be the leading cause of cancer deaths in the UK [1]. In the same year, 62% of patients with lung cancer stages I and IIB and a performance status of 0 to 2 underwent surgery. Around 30% of patients present with locally advanced lung cancer [2]. Currently, patients with locally advanced disease are considered for multimodality treatment if appropriate, according to UK guidelines from the National Institute of Clinical Excellence (NICE) [3]. This includes induction chemotherapy followed by surgery, induction chemoradiotherapy (CRT) followed by surgery, or surgery followed by adjuvant chemotherapy. Multimodality therapy has been shown to improve progression-free survival and may also improve overall survival [3]. Treatment for lung cancer is, however, challenging, as it may involve both local disease and systemic disease control. Appropriate timing and patients’ fitness for treatment are also significant factors that need to be considered. There is extensive discussion regarding the type of modalities of treatment which should be used in locally advanced lung cancer. New developments in targeted therapy and immunotherapy are providing excellent outcomes [4]; but surgery still has a key role in those who have persistent or relapsed disease. In patients with relapsed disease, “salvage” surgery can be offered if a lung primary is considered resectable after chemotherapy or chemoradiotherapy, with acceptable morbidity and mortality and good long-term outcomes. The aim of this paper is to evaluate the role of surgery within multimodality treatment (with radical intent) and as salvage treatment in patients with locally advanced lung cancer, as well to present an analysis of outcome data in our population.

## 2. Materials and Methods

We performed a retrospective analysis of consecutive patients from our institution who underwent surgery after undergoing chemoradiotherapy between 2012 and 2021. There is no consensus definition of salvage surgery in the literature; however, our patients were split into these three groups:Initial plan definitive CRT only: post-treatment imaging showed residual disease deemed to be resectable after discussion at an MDM; therefore, the patient was referred for surgery within 3 months (residual).Initial plan radical treatment with curative intent with trimodality therapy: CRT and surgery (trimodality).Initial plan definitive CRT with a good response initially; however, relapse of disease occurred, and this was deemed resectable, so the patient was referred for surgery. (after at least 3 months) (salvage).

Every patient was discussed at a multidisciplinary meeting (MDM) before being accepted for surgical resection (Appendix A). All patients were managed by the same surgical team. The centre at which patients received CRT was noted. Demographic information, as well as medical history, smoking status, performance status (PS) and lung function tests were recorded. Staging data (using TNM 8th classification) at each point in treatment (pre-CRT, post-CRT and post-surgery), percentage change in size of the primary tumour from PET CT and details of CRT were collected. Details regarding surgery (access, type of resection) and rate of postoperative in-hospital complications, ICU stay and readmission were recorded. 

Relapse-free survival (defined as time from surgical resection to progression of disease), progression-free survival (defined as time from initiation of CRT to progression of disease) and overall survival (defined as the length of time from the date of diagnosis to date of death or last follow up) were calculated and defined in months. 

Statistical analysis was performed with the software Stata version 17. Data are reported as absolute numbers and percentages or as means and standard deviations (SDs), unless otherwise specified. Pearson’s chi-squared test and linear regression modelling (for a comparison of means between more than two groups) and Kaplan–Meier analysis were used for data analysis. Statistical significance was defined as a *p* value < 0.05.

## 3. Results

### 3.1. Patient Characteristics 

A total of 38 patients were included in this study and were split into three groups: residual disease (*n* = 16), trimodality (*n* = 14) and salvage surgery after relapse (*n* = 8). The mean age of patients was 67 years, and the majority of patients were male. The median age was 61 years (47–78 IQR) for the residual disease group, 68 years (55–83 IQR) for the trimodality group and 66 years (56–81 IQR) for the salvage group. 

All subgroup characteristics, including the rate of comorbidities and presence of smoking history, are summarised in Table 1. Overall, there was no significant difference in the demographic data between the groups. There was no significant difference in the lung function between the groups.

### 3.2. Disease Characteristics 

Characteristics of the cancer are shown in Table 2. Tumour location was variable but not significant. In the residual group, 15 were intralobar (crossing the fissure) and 1 tumour was lobar as well as invading the chest wall. In the trimodality group, 1 tumour was hilar, 1 tumour invaded two lobes and the other 12 were intralobar. In the salvage group, 1 tumour was hilar and 7 were intralobar. Pre- and post-CRT, the average percentage reduction in tumour size was 38% in the residual group, 50% in the trimodality group and 43% in the salvage group. There was no significant difference in percentage reduction in size between the groups.

In the trimodality group and salvage surgery group, there was similar staging of disease pre-CRT but the size of tumour pre-CRT in the trimodality group was overall greater (median 5–7 cm) and the size of the tumour in the salvage surgery group (median 4–5 cm). There was no difference in staging between the groups post-treatment. The majority of patients who had no relapse prior to surgery, the majority of these patients had a complete pathological response post-surgery. 

Altogether, 68% (*n* = 26) of the whole cohort initially presented with a minimum of N2 disease. In the residual group, 75% (*n* = 12) of patients had minimum N2 disease pre-CRT. Of these patients, 50% (*n* = 6) downstaged post-CRT, according to PET-CT. A total of 43.8% (*n* = 7) of patients showed downstaging of any nodal disease. 

In the trimodality group, 57.1% (*n* = 8) of patients had N2 disease pre-CRT and 50% (*n* = 4) of these downstaged post-CRT, according to PET-CT. Altogether, 28.6% (*n* = 4) of patients showed downstaging of nodal disease. 

In the salvage group, 75% (*n* = 6) of patients had N2 disease pre-CRT and 16.7% (*n* = 1) downstaged. A total of 33.3% (*n* = 2) of patients showed downstaging of nodal disease. 

Only one patient who had been planned for trimodality therapy had relapse pre-surgery, but this was a distant metastasis, not locoregional relapse.

### 3.3. Treatment

All patients received platinum-doublet chemotherapy either sequentially or concurrently with conformal or IMRT planned external beam radiotherapy. A total of 36 of the 38 patients received a radiotherapy dose equivalent to 60–66 Gy (in 2 Gy per fraction terms). Of the remaining two patients, one had treatment stopped early electively to facilitate surgery and the other received high-dose palliative treatment. Figure 1 describes the CRT regimen given to our patients.

In the residual group, nine (56.3%) patients underwent CRT at their local hospital. One patient received treatment at both their local and at our tertiary referral centre (6.3%). Six patients underwent CRT at our tertiary referral centre (37.5%). In the trimodality group, four patients received CRT at their local hospital (28.6%). Ten patients underwent CRT at our tertiary referral centre (71.4%) and in the salvage group, four patients (50%) underwent CRT at their local hospital and four patients (50%) underwent CRT at our tertiary referral centre.

Average time from initiating chemoradiotherapy to surgery in the residual group was 156 days (range 62–411 days). In the trimodality group, the average time was 107 days (range 38–160 days). In the salvage group, the average time was 405 days (range 227–798 days). The type and regime of CRT given to our patients are described in Figure 1.

Table 3 shows data about surgery. Most patients underwent lobectomy, and most operations were performed by thoracotomy. Most patients were also found to have a complete pathological response post-surgery on histological staging. Altogether, 100% of tumours had an R0 resection. There was no statistically significant difference between the groups in these parameters.

Table 4 shows data about the postoperative course. Postoperative complications were counted if the case ranked class II or above on the Clavien–Dindo classification [5]. The overall complication rate in our patients was 44.7%. The complication rate in the salvage group was 50%. There was no statistically significant difference between the three groups in the rate of hospital complications (including bronchopleural fistula), estimated blood loss, operative time, ICU stay or readmission. 

The postoperative complications in the residual group included pulmonary complications, urinary tract infection, acute kidney injury and atrial fibrillation. The postoperative complications in the trimodality group included lower respiratory tract infections, lung collapse, atrial fibrillation and an intraoperative death. The postoperative complications in the salvage group included pulmonary complications.

### 3.4. Relapse-Free Survival, Progression-Free Survival and Overall Survival

The median overall survival of the study population was 35.8 months (Figure 2). The median overall survival in the residual group was 34.2 months (18.5–61.0 interquartile range). The median overall survival in the trimodality group was 35.4 months (20.1–51.7 IQR). The median overall survival in the salvage group was 35.8 months (32.4–52.7 IQR). There was no statistically significant difference using Kaplan–Meier analysis with a log rank of 0.63. 

In the residual group, 30-day mortality was 0% and 1-year survival was 81.3%. In the trimodality group, 30-day mortality was 7% (*n* = 1) and 1-year survival was 78.6%. In the salvage surgery group, 30-day mortality was 0% and 1-year survival was 62.5%.

Local and distant relapse post-surgery was seen in five patients (31.3%) in the residual group. There was relapse post-surgery in eight patients (57.1%) in the trimodality group and in four patients (50%) in the salvage surgery group. There was no statistically significant difference between the three groups, with a *p* value of 0.34.

The median relapse-free survival was 49 months (19–60 IQR) in the residual group, 33 months (5–53 IQR) in the trimodality group and 18 months (8–36 IQR) in the salvage group (Figure 3).

The median progression-free survival was 26.8 months (9.1–63.8 IQR) in the five patients in the residual group and 11.6 months (7.0–48.3 IQR) in the trimodality group, and the time between initiating CRT and relapse pre-surgery was 7 months in the salvage group (4.25–11 IQR). The median progression-free survival between surgery and post-surgery relapse was 8.1 months (4.6–16.2 IQR).

The 1-year relapse rate was 18.8% in the residual group, 28.6% in the trimodality group and 37.5% (*p* = 0.60) in the salvage group.

## 4. Discussion

The optimal treatment modality or combination therapy for patients with locally advanced lung cancer has not been defined, and surgery has shown good results in previous surgical series published with selected patients [6,7,8,9]. In our series, surgery in combination with CRT either as trimodality treatment or as salvage treatment after relapse obtained a good PFS and overall survival in selected patients.

In our series, there was no significant difference in patient characteristics between the three groups; most patients had a performance status of 1 with good baseline lung function. Generally, patients selected to undergo multimodality treatment tend to be younger and fitter as they must be able to tolerate the rigorous treatment that CRT and surgery involve. 

NICE guidelines from 2019 suggest that trimodality therapy is more effective than CRT alone in those who are fit for surgery and when the cancer is operable [2]. Although the residual disease group in our series was not initially offered surgical resection, they did eventually undergo surgery after re-discussion at the MDM; as our patients were deemed fit enough for surgery. There is selection bias in all our groups, as ethically, patients who are not fit enough and are likely to have poor outcomes are excluded. The decision-making process for the selection was not defined in our study. Selection of patients taking into account their fitness, will be a factor in these studies. In current practice, despite the guidelines, trimodality therapy is not often offered.

Of note, N2 disease (mediastinal disease ipsilateral side to the primary tumour) is considered to be the most important prognostic factor in those with potentially resectable lung cancer. This is because this subset of patients is more likely to develop local and distant relapse [10]. Our groups had no significant difference in the rate of N2 disease, and a relatively similar rate of downstaging of mediastinal disease after CRT compared to other studies has also shown, with rates ranging between 16.7 and 50% in the three groups [11,12,13,14,15]. 

It seems (from the literature) that patients without nodal downstaging and no progression of disease post-CRT seem to have better outcomes with surgery than with no treatment [10]. Others have suggested that after surgical resection, the survival rate is maintained in patients with persistent N2 disease who have undergone definitive CRT; this suggests surgery may have acceptable outcomes despite nodal disease or lack of downstaging. This is supported by our series, which has shown good median overall survival, despite a lack of 100% downstaging of N2 disease. These arguments would indicate in practice that patients with non-bulky N2 disease can feasibly be offered surgery, and those with appropriate operable and resectable cancers should be offered trimodality therapy to begin with, rather than waiting for a downstaging of disease before offering surgical resection.

The length of time from beginning CRT to surgery was longer in the salvage group compared to the trimodality group. This is naturally expected, as surgery was part of the initial management plan for the trimodality patients. The patients in the salvage group would have had a lag in treatment, with a period of follow up before relapse, before a decision was made for surgery. This longer lag time could be associated with the lower overall survival and 1-year survival in the salvage group compared to the trimodality group. Importantly, however, there was no statistically significant difference in overall survival between our groups despite the longer lag time, which leads us to believe that surgery can therefore be considered an acceptable salvage treatment. 

Surgery after radiotherapy is more technically difficult due to radiation-induced fibrosis and may become even more complex with a longer lag time, as is seen in the salvage patients. These patients are therefore less likely to be planned for and undergo minimally invasive and lung-sparing operations. They are also therefore more likely to experience short- and longer-term complications, with other studies recommending that delaying surgery is not recommended due to increased risks of fragile tissue, fistulae formation and bleeding [6,15]. Regarding surgical resection, most of our patients underwent at minimum a lobectomy, with very few undergoing a sublobar resection. This is comparable to a review of 11 case series looking at salvage surgery, where the vast majority underwent lobectomy or pneumonectomy [8]. Our series showed no statistically significant difference between the groups in terms of the in-hospital complication rate, ICU stay, readmission rate or overall survival. Interestingly, there are studies with conflicting suggestions regarding time from RT to salvage surgery. Casiraghi et al. report that a longer time from RT to salvage surgery is associated with longer overall survival; a longer relapse-free interval could indicate a less aggressive cancer, meaning this parameter could be prognostic [7]. Sonobe et al., however, have shown no significant association between time from RT to salvage surgery and patient survival [16]. Our overall complication rate was 28.9% and 37.5% in the salvage group, which is better or comparable to those seen in other studies, which include 25.7% [7], 29.2% [16], 41% [17], 43% [18], 48.4% [19] and 58% [6] after salvage surgery. Our overall rate of postoperative ICU stay was 18.4%, with the salvage group achieving 14.3%, which is again better than reported by other studies, e.g., 48% [6] and 60% [20] after salvage surgery.

NICE guidelines also suggest that the primary advantage associated with chemoradiotherapy and surgery versus chemotherapy alone is the longer progression-free survival time and overall survival [2]. Relapse after treatment is an indicator of poor prognosis, with a median survival of less than 1 year often seen in these patients [2]. The median progression-free survival was longer in the residual disease group compared to the trimodality and salvage groups, though there was no significant difference in patient or tumour characteristics between the groups. Our median overall survival was 36.5 months and 21 months in the salvage group. In comparable salvage groups, the median overall survival ranged widely: 9 months [18], 13 months [7], 13–76 months [9], 30 months [6], 32.5 months [19] and 46 months [20]. Conducting further studies to explore if any particular patient or tumour characteristics are associated with the outcome could help determine who should be offered trimodality therapy in the first instance and those who could safely be offered salvage surgery.

With the establishment of immunotherapy, multimodality treatment involving immunotherapy and surgery is another emerging option for locally advanced NSCLC. Notwithstanding the fact that there are limited data available in the literature with this treatment strategy, the results appear comparable. In a study of 11 patients with immunotherapy as part of their initial treatment and followed by salvage surgery, morbidity was 27%, though no median was OS reported. Their survival at final follow up was 81% (*n* = 9), with a median follow up of 5.9 months [6]. Another study of 38 patients showed a complication rate of 33.4% and median OS of 26.9 months [21].

This study is limited by sample size and it is a single-centre study, so there is inherent selection bias in our patient population, also in terms of fitness and therefore suitability for surgery, as evidenced by the performance status. Due to the retrospective nature of the study, some information, such as the decision making surrounding the initial treatment, was not recorded or not clear. There was also some variation in CRT regimen, though all our patients except one completed the full course given prior to surgery. The CRT was stopped early electively for one patient to facilitate surgery, and another had high-dose palliative treatment. There is no uniform definition of salvage surgery in the current literature, with varied definitions and timings of treatment, which create different populations and therefore make it challenging to compare results directly. Relatively few patients are offered multimodality treatment or salvage surgery, but we have a comparable population number that is even higher than some used in similar studies in this field. Prospective studies over a period of several years and multicentre or international collaboration would be required to have a larger population.

## 5. Conclusions

Multimodality is a recognised treatment for patients with early-stage NSCLC; however, the reported data for this group of patients are limited. We have shown in this retrospective study of 38 patients that multimodality treatment gave impressive survival figures that are in line with those of comparable studies. The downstaging of disease may have led to longer progression-free survival in the residual group compared to the trimodality and salvage surgery groups. There were acceptable or even superior complication rates post-surgery in our population compared to other series, including in salvage surgery. Our study supports the current evidence that multimodality therapy is suitable in early-stage and N2 disease, highlighting particularly that surgery has an important role and can be offered in the first instance to those patients who have operable and resectable lung cancer. Post-CRT, surgery is rarely minimally invasive and sublobar, and given the potential complexity of the management of these patients and the requirement for specialist surgical and critical care input if any complications arise, there should be an emphasis on referral of this patient cohort to centres with experience in surgery post-CRT to establish optimal outcomes.

The landscape for early-stage lung cancer has changed with the arrival of immunotherapy. It is now part of the standard treatment following CRT [22]. Additionally, emerging data from the NADIM trial and others on combined chemotherapy, immunotherapy and surgery look set to change things further, with preliminary evidence suggesting a considerable rate of pathological response after immunotherapy and even better response rates with chemo-immunotherapy [23,24]. We are fortunate at present to have more treatments at our disposal to offer early-stage NSCLC patients, and whilst trying to navigate through the emerging data, it is important to be mindful that trimodality therapy should be considered as a management plan when discussing all treatments for lung cancer patients in the multidisciplinary setting. We have also shown that surgery is a safe and reasonable option for those with locally advanced lung cancer after CRT and those who do end up with relapsed disease. Further study on patient or disease factors that may influence progression-free and other survival parameters and the evaluation of outcomes of those undergoing surgery, particularly with immunotherapy–chemotherapy, would be informative in determining the decision-making process. Prospective studies in this patient population and establishing a universal definition of salvage surgery would also be useful.

## Figures and Tables

**Figure 1 cancers-15-03586-f001:**
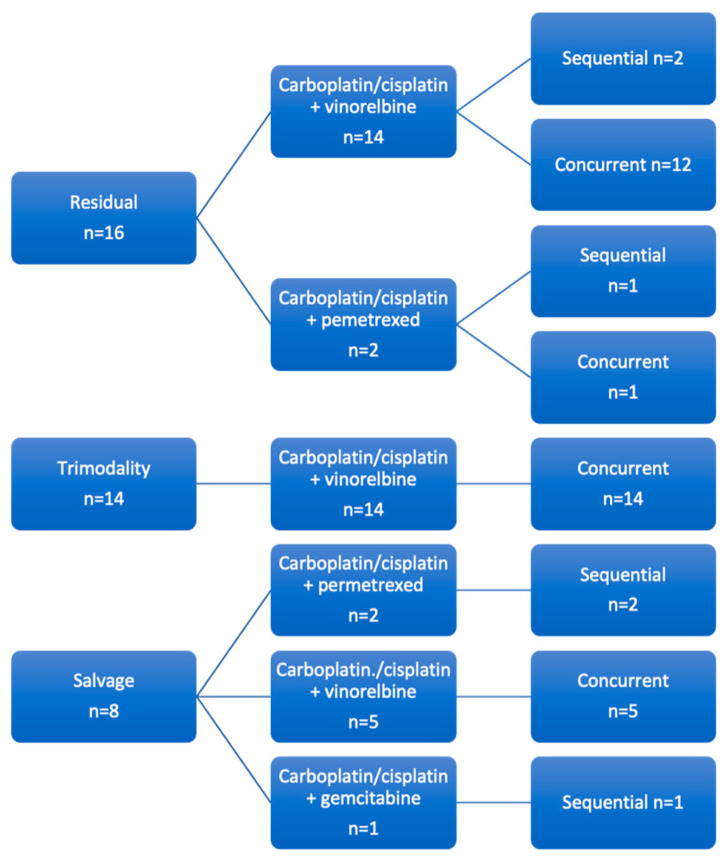
Chemotherapy Regimen. This figure shows the chemotherapy regimen given to our patients, divided into residual, trimodality and salvage groups.

**Figure 2 cancers-15-03586-f002:**
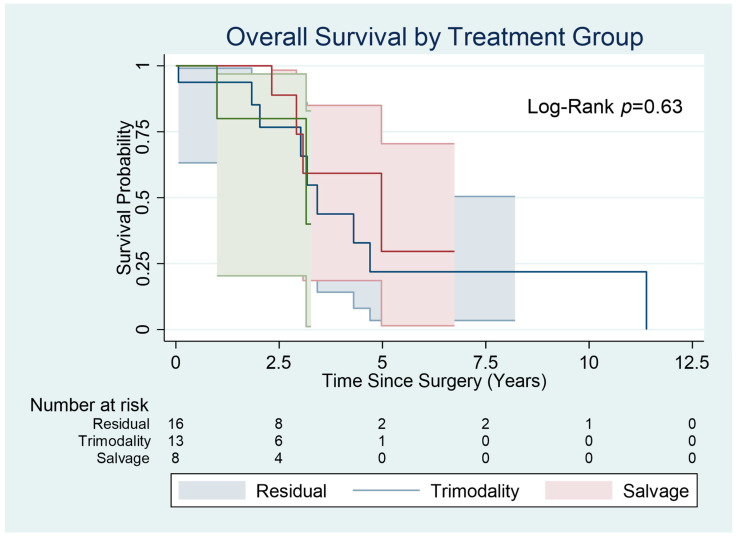
Overall Survival by Treatment Group.

**Figure 3 cancers-15-03586-f003:**
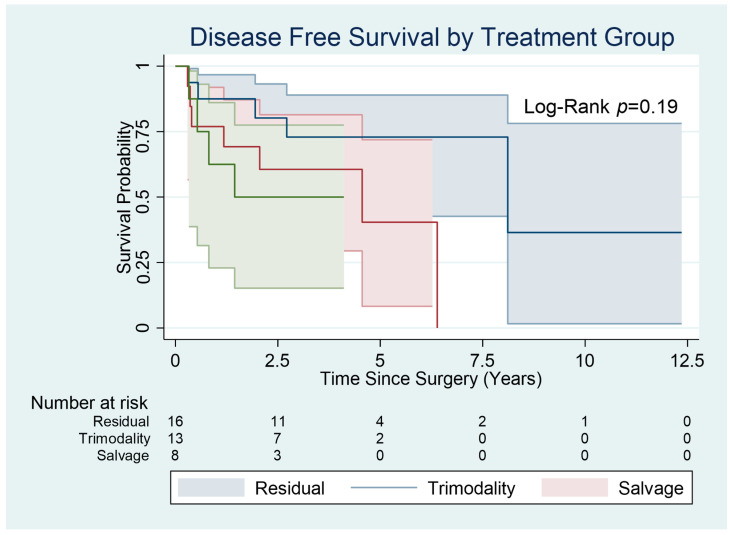
Disease-Free Survival (Relapse-Free Survival) By Treatment Group.

**Table 1 cancers-15-03586-t001:** Demographic Details—Residual, Trimodality and Salvage Groups.

Characteristic	Variables	Residual *n* (%)	Trimodality *n* (%)	Salvage *n* (%)	*p* Value
Sex	Female	6 (37.5)	3 (21.4)	3 (37.5)	0.59
	Male	10 (62.5)	11 (78.6)	5 (62.5)	
Age	<65	6 (37.5)	3 (21.4)	4 (50)	0.14
	≥65	10 (62.5)	11 (78.6)	4 (50)	
Comorbidities	Pulmonary	5(31.3)	4 (28.6)	6 (75)	0.07
	Cardiac	8 (50)	7 (50)	2 (25)	0.47
	Other	9 (56.3)	8 (57.1)	3 (37.5)	0.63
Smoking History	Current	4 (25)	5 (35.7)	2 (25)	0.78
	Ex smoker	12 (75)	9 (64.3)	6 (75)	
	No	0 (0)	0 (0)	0 (0)	
	>20	9 (56.3)	10 (71.4)	4 (50)	
	Unknown	2 (12.5)	2 (14.3)	2 (25)	
ECOG/WHO Performance Status	0	5 (31.3)	7 (50)	0 (0)	0.06
	1	10 (62.5)	7 (50)	5 (62.5)	
	2	1 (6.3)	0 (0)	2 (25)	
	3	0 (0)	0 (0)	1 (12.5)	
		Residual	Trimodality	Salvage	*p* value
		Median (IQR)	Median (IQR)	Median (IQR)	
Lung Function	FEV1 %	82.0 (64.0–93.0)	79 (74–87.9)	77.5 (68.8–88.5)	0.80
	FVC %	100 (87.8–110.8)	96.7 (92.5–119.5)	104.5 (96.8–112.3)	0.50
	Transfer Factor %	80 (58–96)	66 (61–79.0)	67.5 (49,8–76.8)	0.19

This table shows demographic details of the patient split into three groups—residual, trimodality and salvage. Statistical analysis was performed to check for any significant difference in demographic data between the groups.

**Table 2 cancers-15-03586-t002:** Disease Characteristics—Residual, Trimodality and Salvage Groups.

Characteristic	Variables	Residual *n* (%)	Trimodality *n* (%)	Salvage *n* (%)	*p* Value
Stage pre-chemoradiotherapy	1A	0 (0)	0 (0)	0 (0)	0.15
	1B	0 (0)	0 (0)	0 (0)	
	2A	0 (0)	0 (0)	0 (0)	
	2B	0 (0)	2 (14.3)	2 (25)	
	3A	7 (43.8)	8 (57.1)	1 (12.5)	
	3B	7 (43.8)	4 (28.6)	3 (37.5)	
	3C	2 (12.5)	0 (0)	2 (25)	
	4	0 (0)	0 (0)	0 (0)	
Tumour location	Right upper lobe	8 (50)	6 (42.9)	1 (12.5)	0.17
	Right middle lobe	0 (0)	0 (0)	1 (12.5)	
	Right lower lobe	0 (0)	1 (7.1)	0 (0)	
	Left upper lobe	5 (31.3)	4 (28.6)	3 (37.5)	
	Left lower lobe	1 (6.3)	0 (0)	2 (25)	
	Other	2 * (12.5)	1 * (7.1)	1 * (12.5)	
	No residual	0 (0)	1 (7.1)	0 (0)	
Size of tumour pre-chemoradiotherapy	<1 cm	0 (0)	0 (0)	0 (0)	0.13
	1–2 cm	0 (0)	0 (0)	1 (12.5)	
	2–3 cm	1 (6.3)	2 (14.3)	0 (0)	
	3–4 cm	1 (6.3)	0 (0)	3 (37.5)	
	4–5 cm	2 (12.5)	2 (14.3)	0 (0)	
	5–7 cm	7 (43.8)	5 (35.7)	3 (37.5)	
	>7 cm	5 (31.3)	5 (35.7)	1 (12.5)	
Histological type	Non-small cell	16 (100)	14 (100)	8 (100)	
	Adenocarcinoma	11 (68.8)	8 (57.1)	4 (50)	0.64
	Squamous	5 (31.3)	6 (42.9)	4 (50)	
Stage post-chemoradiotherapy	Complete response	0 (0)	1 (7.1)	0 (0)	0.65
	1A	1 (6.3)	1 (7.1)	1 (12.5)	
	1B	6 (37.5)	4 (28.6)	0 (0)	
	2A	0 (0)	2 (14.3)	1 (12.5)	
	2B	2 (12.5)	2 (14.3)	1 (12.5)	
	3A	5 (31.3)	4 (28.6)	4 (50)	
	3B	2 (12.5)	0 (0)	1 (12.5)	
	3C	0 (0)	0 (0)	0 (0)	
	4	0 (0)	0 (0)	0 (0)	
Size of tumour post-chemoradiotherapy	<1 cm	0 (0)	1 (7.1)	0 (0)	0.11
	1–2 cm	3 (18.8)	3 (21.4)	2 (25)	
	2–3 cm	1 (6.3)	2 (14.3)	4 (50)	
	3–4 cm	6 (37.5)	5 (35.7)	2 (25)	
	4–5 cm	2 (12.5)	2 (14.3)	0 (0)	
	5–7 cm	3 (18.8)	1 (7.1)	0 (0)	
	>7 cm	1 (6.3)	0 (0)	0 (0)	

This table shows the disease characteristics by residual, trimodality and salvage groups. Statistical analysis was performed to check for any significant difference in demographic data between the groups. * interlobar.

**Table 3 cancers-15-03586-t003:** Treatment Given and Staging Post-Treatment—Residual, Trimodality and Salvage Groups.

Characteristic	Variable	Residual *n* (%)	Trimodality *n* (%)	Salvage *n* (%)	*p* Value
Surgical resection	Pneumonectomy	1 (6.3)	0 (0)	2 (25)	0.30
	Bilobectomy	0 (0)	1 (7.1)	0 (0)	
	Lobectomy	14	12 (85.7)	6 (75)	
	With chest wall resection	3 (21.4)	3 (25)		
	Segmentectomy	1 (6.3)	0 (0)	0 (0)	
	Other	0 (0)	1 * (7.1)	0 (0)	
Surgical technique	Thoracotomy	14 (87.5)	13 (92.9)	7 (87.5)	0.46
	Hemiclamshell	1 (6.3)	1 (7.1)	0 (0)	
	Minimally invasive	1 ^ (6.3)	0 (0)	0 (0)	
	Converted to open	0 (0)	0 (0)	1 (12.5)	
Stage post-surgery	Complete pathological response	12 (75)	7 (50)	1 (12.5)	0.07
	1A	1 (6.3)	3 (21.4)	1 (12.5)	
	1B	1 (6.3)	0 (0)	2 (25)	
	2A	0 (0)	1 (7.1)	0 (0)	
	2B	1 (6.3)	2 (14.3)	4 (50)	
	3A	1 (6.3)	0 (0)	0 (0)	
	CPR but metastatic	0 (0)	1 (7.1)	0 (0)	
R status	0		14 (100)	8 (100)	0.49
	1		0 (0)	0 (0)	

This table describes the treatment given and staging post-treatment in the residual, trimodality and salvage groups. * Exploratory thoracotomy, intraoperative death, ^ robotic.

**Table 4 cancers-15-03586-t004:** Postoperative Course.

Characteristic	Residual *n* (%)	Trimodality *n* (%)	Salvage *n* (%)	*p* Value
In-hospital complications	5 (31.3)	8 (57.1)	4 (50)	0.82
Bronchopleural fistula	0 (0)	1 (7.1)	0 (0)	
	Residual (millilitres)	Trimodality (millilitres)	Salvage (millilitres)	
Estimated blood loss	971.4 (2049.7)	842.3 (1611.3)	325.0 (331.7)	0.28
	Residual mean minutes (SD)	Trimodality minutes (SD)	Salvage minutes (SD)	
Operative time	188.1 (52.3)	182.5 (78)	146 (32.4)	0.80
ICU stay	5 (31.3)	1 (7.1)	1 (12.5)	0.21
Readmission within 30 days	0 (0)	0 (0)	0 (0)	1
	Residual (mean days)	Trimodality (mean days)	Salvage (mean days)	*p* value
Total length of stay	12.8	9.7	18.5	0.087

## Data Availability

Data available on request due to ethical restrictions. The data presented in this study are available on request from the corresponding author. The data are not publicly available due to patient confidentiality.

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
