# Peer review of "Multimodality Treatment and Salvage Surgery for the Treatment of Lung Cancer"

_cancers, 2023, doi:10.3390/cancers15143586_

Round 1

Reviewer 1 Report

This manuscript explored patients who underwent multimodality therapy. Generally well written. However, the single-institution study of 38 patients was very small in number, subject to selection bias, and varied in chemotherapy regimens. As a result, there were few new discoveries for the reader.

1. The lung function in Figure 1 should use the median rather than the mean because the population is probably not normally distributed.

2. References must be numbered in order of appearance in the text. In the text, reference numbers should be placed in square brackets [ ]. 

Your references not be numbered in order of appearance in the text. The references must be sorted correctly. Also, some references were not enclosed in square brackets [ ]. → Line 184, 256.

Author Response

Many thanks for your time in reviewing our paper.

Point 1: This manuscript explored patients who underwent multimodality therapy. Generally well written. However, the single-institution study of 38 patients was very small in number, subject to selection bias, and varied in chemotherapy regimens. As a result, there were few new discoveries for the reader.

Response 1: We have performed a preliminary retrospective study representing the practice at our institution on the multimodality treatment of lung cancer patients. Based on our experience, relatively few patients are offered multimodality treatment or salvage surgery, so the sample is likely to be small even over a period of several years. We have a comparable population number or even larger to similar studies in this field [1]. Despite the small sample size, the population reflects real world data and the complexity of this patient population. We have recognised the limitations and have commented on them in our discussion. In order to have a larger population, it is likely that a prospective study over a period of several more years, multi-centre or even international collaboration would be required. This is mentioned in our paper (lines 319-332) We feel that this is an interesting institutional study and there remain useful insights and suggestions for further study in this specific population. We have shown there can be good outcomes in this population and a prospective multi-centre study may show even better survival and complication rates.

  1. Hamada, A., Soh, J. and Mitsudomi, T., 2021. Salvage surgery after definitive chemoradiotherapy for patients with non-small cell lung cancer. Translational Lung Cancer Research, 10(1), p.555.

Point 2: The lung function in Figure 1 should use the median rather than the mean because the population is probably not normally distributed.

Response 2: The lung function data has been changed to median (IQR), please see Table 1.

Point 3. References must be numbered in order of appearance in the text. In the text, reference numbers should be placed in square brackets [ ]. 

Response 3: The references have been changed to be numbered in order of appearance.

Point 4: Your references not be numbered in order of appearance in the text. The references must be sorted correctly. Also, some references were not enclosed in square brackets [ ]. → Line 184, 256.

Response 4: The reference have been changed to be numbered in order of appearance and the citations have been checked so they are all within square brackets.

Reviewer 2 Report

Authors have recognized well the limitations of the study including small sample size and the retrospective nature of the study. In spite of these limitations, it is a nice institutional review of clinical importance.  

Author Response

Point 1: Authors have recognized well the limitations of the study including small sample size and the retrospective nature of the study. In spite of these limitations, it is a nice institutional review of clinical importance.  

Response 1: Many thanks for your very kind review.

Reviewer 3 Report

The authors analyze the role of surgery in the multimodal treatment of lung cancer, both as a curative procedure and as a salvage treatment. This is a monocentric retrospective study on 38 patients (main limitation) with locally advanced lung cancer undergoing surgical treatment after CRT; patients are divided into 3 groups, according to the reason for CRT: programmed surgery after CRT (trimodal treatment); residual resectable disease; salvage surgery in local recurrence after CRT. In all groups downstaging of N2-disease was significant after CRT, and the negativity of other nodal stations after CRT was very important. Another important aspect is that more than 50% of the patients received CRT in local hospitals instead of the reference center for thoracic surgery. The three groups do not show significance in the presentation of the disease, in the type of surgery performed, in complications, and in survival. All patients had a good performance status; in the study, data from patients with a low-performance status were not reported as they were excluded based on this criterion. I agree that in patients with non-bulky N2 resectable disease, surgery may be offered as part of the treatment. I also agree that post-CRT surgery is rarely minimally invasive and sublobar. The authors bring attention to the role of surgery in locally-advanced lung cancer after CRT, and this could motivate in the future the role of surgery after immunotherapy-CHT. Moreover, a delayed surgery could reduce the complication of extensive surgery (doi: 10.21037/jtd.2018.07.21). I think this article is rich in new insights about this topic and introduce question on other types of treatment for cancer that is of paramount interest in the field of oncology (doi: 10.3390/curroncol29100538).

Author Response

Point 1: The authors analyze the role of surgery in the multimodal treatment of lung cancer, both as a curative procedure and as a salvage treatment. This is a monocentric retrospective study on 38 patients (main limitation) with locally advanced lung cancer undergoing surgical treatment after CRT; patients are divided into 3 groups, according to the reason for CRT: programmed surgery after CRT (trimodal treatment); residual resectable disease; salvage surgery in local recurrence after CRT. In all groups downstaging of N2-disease was significant after CRT, and the negativity of other nodal stations after CRT was very important. Another important aspect is that more than 50% of the patients received CRT in local hospitals instead of the reference center for thoracic surgery. The three groups do not show significance in the presentation of the disease, in the type of surgery performed, in complications, and in survival. All patients had a good performance status; in the study, data from patients with a low-performance status were not reported as they were excluded based on this criterion. I agree that in patients with non-bulky N2 resectable disease, surgery may be offered as part of the treatment. I also agree that post-CRT surgery is rarely minimally invasive and sublobar. The authors bring attention to the role of surgery in locally-advanced lung cancer after CRT, and this could motivate in the future the role of surgery after immunotherapy-CHT. Moreover, a delayed surgery could reduce the complication of extensive surgery (doi: 10.21037/jtd.2018.07.21). I think this article is rich in new insights about this topic and introduce question on other types of treatment for cancer that is of paramount interest in the field of oncology (doi: 10.3390/curroncol29100538).

Response 1: Many thanks for your kind review and your time in reviewing our paper. We agree with your points and have made some changes to our conclusions lines 336-348 and lines 358-364..

Reviewer 4 Report

It’s well  known that locally advanced lung cancer patients are very heterogeneous and only highly selected patients could benefit from surgery after CRT. The author conducted a retrospective study of these patients treated with CRT plus Surgery after dividing them into three groups. So more details of the criterion for each group (line 60-64)should be provided。It’ll  be better if each patient’s MDT meeting information including preCRT TNM stage, postCRT TNM stage, staging tools, and the reasons for surgery instead of other alternative treatments could be attached in supplementary.  

line 183-196 and table 4, suggest the author add data about broncho pleural fistula complications  and the extent of surgical difficulties indicated by blood loss and operative time, which are the two most concerning problems for surgeons when operating CRT patients.

Author Response

Many thanks for your time in reviewing our paper.

Point 1: It’s well known that locally advanced lung cancer patients are very heterogeneous and only highly selected patients could benefit from surgery after CRT. The author conducted a retrospective study of these patients treated with CRT plus Surgery after dividing them into three groups. So more

details of the criterion for each group (line 60-64) should be provided。It’ll  be better if each patient’s MDT meeting information including preCRT TNM stage, postCRT TNM stage, staging tools, and the reasons for surgery instead of other alternative treatments could be attached in supplementary.

Response 1: The criterion for each has been updated in the methods section (lines 60-67). The MDM information for each patient is available in Appendix A, along with the preCRT TNM stage, postCRT TNM stage and the reason for surgery instead of alternative treatments. The staging tool is the TNM 8th classification is mentioned in the methods (line 72).

Point 2: line 183-196 and table 4, suggest the author add data about broncho pleural fistula complications  and the extent of surgical difficulties indicated by blood loss and operative time, which are the two most concerning problems for surgeons when operating CRT patients.

Response 2: We agree these data points are provide useful information. The bronchopleural fistula complication, blood loss and operative time have been added to Table 4. A comment regarding no significant difference between the groups is made in lines 190-191.

Round 2

Reviewer 1 Report

The part I pointed out has been corrected.